

# Undercarboxylated osteocalcin inhibits the early differentiation of osteoclast mediated by Gprc6a

Hailong Wang[1,*], Jinqiao Li[1,2,*], Zihan Xu[1], Feng Wu[1], Hongyu Zhang[1], Chao Yang[1], Jian Chen[1,3], Bai Ding[1], Xiukun Sui[1], Zhifeng Guo[1], Yinghui Li[1] and Zhongquan Dai[1]

[1] State Key Laboratory of Space Medicine Fundamentals and Application, China Astronaut Research and Training Center, Beijing, China
[2] Space Engineering University, Beijing, China
[3] Department of Orthopaedics, Sir Run Run Shaw Hospital, School of Medicine, Zhejiang University, Hangzhou, China
[*] These authors contributed equally to this work.

Corresponding authors
Yinghui Li, yinghuidd@vip.sina.com
Zhongquan Dai, daizhq77@163.com

## ABSTRACT

Osteocalcin (OCN) was the most abundant noncollagen protein and considered as an endocrine factor. However, the functions of Undercarboxylated osteocalcin (ucOCN) on osteoclast and bone resorption are not well understood. In the present study, preosteoclast RAW264.7 cells and bone marrow mononuclear cells (BMMs) were treated with ucOCN purified from prokaryotic bacteria. Our results showed that ucOCN attenuated the proliferation of RAW264.7 cells with a concentration dependant manner by MTS assay. Scrape wounding assay revealed the decreased motility of RAW264.7 cells after ucOCN treatment. RT-qPCR results manifested the inhibitory effects of ucOCN on the expression of osteoclastic marker genes in RAW264.7 cells during inducing differentiation of RANKL. It was also observed that ucOCN inhibited the formation of multinucleated cells from RAW264.7 cells and BMMs detected by TRAP staining. The number and area of bone resorb pits were also decreased after treatment with ucOCN during their osteoclast induction by toluidine blue staining. The formation and integrity of the osteoclast actin ring were impaired by ucOCN by immunofluorescent staining. Time dependant treatment of ucOCN during osteoclastic induction demonstrated the inhibitory effects mainly occurred at the early stage of osteoclastogenesis. Signaling analysis of luciferase activity of the CRE or SRE reporter and ERK1/2 phosphorylation showed the selective inhibitor or siRNA of Gprc6a (a presumptive ucOCN receptor) could attenuate the promotion of ucOCN on CRE-luciferase activity. Taken together, we provided the first evidence that ucOCN had negative effects on the early differentiation and bone resorption of osteoclasts via Gprc6a.

## INTRODUCTION

Osteocalcin (OCN) is one of the most abundant noncollagen proteins and initially considered as an osteoblast-specifically secreted biomarker of bone turnover for the clinical diagnosis (*Kruse & Kracht, 1986*). OCN protein consists of 46–50 amino acids according to the species, and contains three highly conserved glutamic acid residues (Glu), which were carboxylated by a vitamin K dependent $\gamma$-glutamyl carboxylase (GGCX) (*Hauschka et al., 1989*). The carboxylated osteocalcin (cOCN) has a stronger affinity with hydroxyapatite (HA) in the presence of calcium ions and then deposits into the bone matrix (*Hauschka & Carr, 1982*; *Li et al., 2016*). OCN is thought to play a vital role in skeletal development. Previous studies have showed that the percentage of undercarboxylated OCN (ucOCN) was increased early and remained high during spaceflight, which means OCN maybe play an important role during bone alternation during microgravity exposure (*Caillot-Augusseau et al., 2000*). The exact function of OCN in bone and the whole body is still unknown (*Diegel et al., 2020*).

Although there were some controversial data from in vitro and in vivo studies, the dual roles of OCN in bone were presumed: (1) modulating the activity of osteoblasts and osteoclasts, (2) acting as a regulator of bone mineralization (*Neve, Corrado & Cantatore, 2013*). The expression of OCN increased with the later stage of osteoblast differentiation and highly expressed in mature osteoblasts and preosteocytes. Contrary to expectations, OCN-deficient mice developed a phenotype characterized with higher bone mass and biomechanical properteries, but without impaired bone resorption (*Ducy et al., 1996*). Completed knockout of OCN resulted in significantly increased trabecular thickness, density, and volume in rat bone (*Lambert et al., 2016*). Overexpressing OCN in mice bone showed a relatively normal state of mineralization (*Murshed et al., 2004*). The exact function of OCN in bone homeostasis remains controversial.

Involvement of OCN in bone resorption was also investigated, especially in the regulation of osteoclast recruitment, differentiation, and activity. Purified OCN from lyophilized chicken bone promoted a dose dependent chemotactic response in osteoclast progenitor monocytes (*Malone et al., 1982*). The bone particles (BP) from warfarin treated rat (daily subcutaneous injections of sodium warfarin 1 day after birth for 42 days) showed little OCN and less Gla content than that from normal rats. After implantation into bilateral, subcutaneous pockets of normal rat, the warfarin treated BPs were resorbed only 60% of normal BP detected by histomorphometric analysis and recruited fewer osteoclasts with decreased tartrate-resistant acid phosphatase (TRAP) activity (*Glowacki & Lian, 1987*; *Lian, Tassinari & Glowacki, 1984*; *Liggett et al., 1994*). These results suggested that OCN deficiency could inhibit the recruitment of osteoclast progenitor and its differentiation, especially carboxylated OCN, and OCN played an important role in the osteoclast development and functions. However, genetic studies demonstrated deletion of OCN resulted in unimpaired bone resorption after ovariectomy (*Ducy et al., 1996*). Due to the decreased GGCX activity and the low pH during bone resorption, some unincorporated cOCN and undercarboxylated OCN (ucOCN) are released into the circulation and considered as a clinical marker of bone turnover (*Clemens & Karsenty, 2011*; *Plantalech et*

*al., 1991*) and have a concentration range of 40–228 ng/ml in mouse serum (*Srivastava et al., 2000*). Recently, increasing evidences indicate that ucOCN is an endocrine hormone which regulates energy metabolism (*Lee et al., 2007*), male fertility (*Oury et al., 2011*), muscle mass (*Mera et al., 2016*), brain development, and recognition (*Oury et al., 2013*). But the exact function and regulation model of ucOCN in bone are far away understood, especially in osteoclast and bone resorption (*Manolagas, 2020*). Furthermore, Gprc6a, a G protein-coupled receptor class C group 6, was considered as its candidate receptor (*Oury et al., 2011*), but not all thought so (*Rueda et al., 2016*).

In the present study, we hypothesized that ucOCN could affect the maturity and function of osteoclasts. ucOCN was purified from artificially constructed prokaryotic bacteria and added to preosteoclast RAW264.7 cells and BMMs. The results showed that ucOCN decreased the proliferation, motility, differentiation, and maturation of the preosteoclast. Further reporter assay and knockdown examination demonstrated the Gprc6a mediated the negative regulation of ucOCN on osteoclast.

## MATERIALS AND METHODS

### Cells cultures

All the cell lines used in the present study were obtained from Chinese National Infrastructure of Cell Line Resource. RAW264.7 cells were cultured in DMEM (HyClone, USA) supplemented with 10% FBS (Gibco, USA) and incubated at 37 °C in 5% $CO_2$ air humidified incubator. BMMs were isolated from the long bones of 6-week-old mice and maintained in $\alpha$-minimal essential medium containing 10% FBS in the presence of MCSF (10 ng/ml, Pepro Tech., USA). To generate osteoclasts from BMMs, cells were plated in 24-well tissue culture plates and cultured in the presence of 30 ng/ml MCSF and 50 ng/ml RANKL (R&D System, USA).

### The ucOCN production

GST-OCN fusion protein was bacterially produced and purified by Glutathione Sepharose Fast Flow column (GE Healthy Life Science, USA) according to standard procedures. OCN protein/gene was then cleaved out from the GST subunit using thrombin. Its purity and concentration were evaluated by ELISA (Clontech, USA).

### Cell migration assays

Migration was measured by wound healing assay. RAW264.7 cells were plated at a density of $4 \times 10^5$ cells/ml into 24-well plates. When cells achieved 80% confluence, the plates were streaked with a sterile micropipette tip. After rinsing off the released cells, the plates were visualized at $10 \times$ magnification along with the streaks at time 0 and 24 h later. The cell migration was determined by measuring wound width (in pixels) using ImageJ software (NIH, USA).

### Cell proliferation analysis by MTS

RAW264.7 cells were seeded into a 96-well plate at a density of $1 \times 10^5$ cells/well, and replaced medium with different concentration of ucOCN (1, 10, 100, 1,000 ng/ml) after
**Table 1  The primers list.**

| Gene name | Forward | Reverse |
| --- | --- | --- |
| c-Fms | TCTTACGCAAAACGGTCTACTTC | CCAATTTTATCTGTGGGGGC |
| RANK | TTGCTTCCCTGCTGGATTAG | AAGACGGTGCTGGAGTCTGT |
| TRAF6 | TTGCACATTCAGTGTTTTTGG | TGCAAGTGTCGTGCCAAG |
| NFATc1 | AGATACCACCTTTCCGCAAC | TAATTGGAACATTGGCAGGA |
| TRAP | GCGACCATTGTTAGCCACATACG | CGTTGATGTCGCACAGAGGGAT |
| CathK | GCGTTGTTCTTATTCCGAGC | CAGCAGAGGTGTGTACTATG |
| Tcirg1 | CCATATCCCTTTGGCATTGA | GAGAAAGCTCAGGTGGTTCG |
| CLC7 | GTCCTTCAGCCTCAGTCG | ACACAGCGTCTAATCACAAC |
| MMP9 | GCTGACTACGATAAGGACGGCA | GCGGCCCTCAAAGATGAACGG |
| CTR | TGGTGCGGCGGGATCCTATAAGT | AGCGTAGGCGTTGCTCGTCG |
| GPRC6A | GCTCGAGACTGCAAGAAACC | TGAAGGCCAGAACTGTGATG |
| GAPDH | ACTCCACTCACGGCAAATTCA | GGCCTCACCCCATTTGATG |

adhesion about 24 h. Then, the effect of different concentrations of ucOCN on RAW264.7 cell proliferation was analyzed using the MTS assay kit according to the instructions of the manufacturer (Promega, USA).

## RT-qPCR for gene expression analysis

Total RNAs were extracted using the Trizol reagent according to the standard protocol. The cDNA was synthesized from 500 ng of total RNA using PrimeScript[TM] RT reagent Kit with gDNA Eraser (Takara, Dalian China) according to the manufacturer's instructions, and performed the analysis of gene expression using TB Green[TM] Ex Taq[TM] II (Tli RnaseH) from Takara on a Roche lightCycler 96 instrument with 0.4 µM gene-specific primer pairs (Table 1) respectively; GAPDH amplification was used as an internal reference for each sample.

## TRAP staining

RAW264.7 cells were seeded in a 6-well plate with 100 ng/ml ucOCN in the presence of 50 ng/ml RANKL. BMMs were treated with ucOCN (100 ng/ml) in the presence of 30 ng/ml MCSF and 50 ng/ml RANKL. The medium was replaced every alternate day. After 4 or 6 days of culture, TRAP histochemical staining was performed using a leukocyte acid phosphatase kit (Sigma,USA) following the instruction.

## Toluidine blue staining of bovine bone slices for osteoclast resorption

Toluidine blue was a basic thiazine metachromatic dye with high affinity for acidic tissue components and widely utilized for bone resorption pit staining. In brief, bovine slices (Beijing Keruimei technology CO. LTD) were dipped in PBS and medium with 10-fold high concentration of streptomycin and penicillin for 12 h, respectively. RAW264.7 cells or BMMs were seeded onto the slices at a density of $1 \times 10^5$ cells/well with the inducing factors or coupled with ucOCN treatment. The medium was replaced every alternate day. After 10 days of culture, the slices were washed with PBS followed by fixation with 4%

paraformaldehyde for 20 min. After this, cells were washed twice with PBS and removed from bone slices via ultrasonication in 0.25 M $NH_4OH$ for 3 times followed by distilled water. Resorption pit formation was stained by 0.1% toluidine blue (Amersco, USA) for 10 min at room temperature. The slices were then rinsed with distilled water more than 5 times to excluderesidues. Resorption pits are now stained in dark blue and images were taken via light microscopy.

## Actin ring staining

Actin rings of mature osteoclasts were detected by actin filaments staining with rhodamine-conjugated phalloidin (molecular probe, USA). Mature osteoclasts were formed from BMMs cultures in the presence of RANKL (50 ng/ml) and MSCF (30ng/ml). At the end of incubation with ucOCN treatment for 6 days, osteoclasts were stained with rhodamine-conjugated phalloidin for actin. The distribution of actin rings was visualized and the images were taken by a Nikon Eclipse Ti microscope (Japan).

## Western blot analysis

The cells were lysed by RIPA buffer (50 mM Tris–HCl (pH 7.4), 150 mM NaCl, 1% NP-40, 1 mM EDTA, and 0.1% sodium dodecyl sulfate) containing protease inhibitor cocktail (Roche, German). Each sample was kept on ice for 30 min before it was centrifuged at 13,000 g for 30 min at 4 °C. The protein concentration of the supernatant was determined with a bicinchoninic acid protein assay kit (Thermo Fisher Scientific, USA). The total proteins were subjected to SDS-PAGE and then transferred to a polyvinylidene difluoride membrane (Millipore, USA). The membranes were blocked with 5% skim milk in Tris-buffered saline containing 0.1% Tween 20 (TBST) for 1 h and then incubated overnight with a primary antibody diluted in 5% bovine serum albumin at 4 °C (specific antibodies for RANK, c-Fms (Abcam, UK), P44/42 MAPK(REK1/2), Phospho-p44/42 MAPK(ERK1/2), GAPDH (CST, USA), Grpc6a (Santa Cruz, USA)). The membranes were incubated with appropriate secondary antibodies conjugated with horseradish peroxidase (CST, USA) for 2 h at room temperature. Signals were detected using enhanced luminescence (Bio-Rad, USA) after washing three times. The intensity of the protein bands was calculated using ImageJ software (NIH, USA).

## Luciferase reporter gene assay

RAW264.7 cells were seeded into 24-well plates at a density of with $1.5 \times 10^5$ cells/ml and cultured for 12 h. The specific siRNA of Gprc6a (siRNA-2553 GCAGAAGACTAA-CACCAAA) was transfected into RAW264.7 cells for 24 h before reporter transfection and ucOCN treatment. The reporter plasmid pGL4.33[luc2P/SRE/Hygro] or pGL4.29 [luc2P/CRE/ Hygro] coupled with phRLTK was transfected into RAW264.7 cells followed by treatment with 100 ng/ml ucOCN, 20 mM L-ornithine (Sigma, USA) alone or combined with 30 μM NPS2143 (Sigma, USA) as previous reports desrption (*Rueda et al., 2016*) 36 h later. The measurement of luciferase activity was performed with the Dual Luciferase Reporter Assay kit (Promega, USA) following the manufactory's instruction.

## Statistical analysis

Experiments were carried out independently at least three times. Results are expressed as the mean ± SEM and were compared by Student's t test (two groups) or Two-way repearted measures ANOVA with Duncan's multiple comparison (multiple groups). Results were considered significantly different for $p < 0.05$.

## RESULTS

Osteoclast precursors are attracted from the bone marrow to the bloodstream by chemokines and recruited to bone by a variety of factors released at sites undergoing resorption and differentiated into mature osteoclasts to resorb bone tissue (*Boskey et al., 2003*). Previous studies have demonstrated that OCN could recruit and promote osteoclast differentiation, but the mechanism was still not well understood, especially ucOCN. Thus, we tried to evaluate the effects of ucOCN on the migration, proliferation, and differentiation of osteoclast recruitment process using RAW264.7 cells.

### ucOCN attenuated the proliferation and migration of RAW264.7 cells

First of all, as the previous report (*Oury et al., 2011*), we purified ucOCN from an artificial E.coli prokaryotic bacteria by GST-fusion method. Its purity and concentration were examined by specific ELISA kit. As shown in Fig. S1, no cOCN was detected by specific ELISA kit and a very low of endotoxin level was detected. The purified ucOCN was used in the following experiments.

To explore the effects of ucOCN on osteoclast proliferation, the mouse monocyte RAW264.7 cells were treated with different concentration of ucOCN (1, 10, 100, 1,000 ng/ml) for 24 h, 48 h and 72 h. The growth rate of the different treated cells was detected by MTS method and it was significantly inhibited by ucOCN treatment at the indicated time point as shown in Fig. 1A. The inhibition effect of ucOCN at the lowest concentration (1 ng/ml) just lasted for 48 h, but the inhibition effects of ucOCN at other three concentrations (10, 100, 1,000 ng/ml) still significantly occurred at 72 h time point. There was about 40% inhibition at 24 h and 48 h, but only 20–30% at 72 h. These results indicated that ucOCN attenuated the proliferation activity of RAW264.7 cells with concentration and time-dependant pattern.

The migration of RAW264.7 cells was measured in a scrape wounding assay over 24 h with ucOCN (100 ng/ml) treatment or not. As shown in Figs. 1B–1C, there was a higher motility in untreated RAW264.7 cells than ucOCN treated cells. In the control group, the cells at the edge of the steak were significantly migrated toward the middle and the width of the scribe line became narrowed. But there were almost no changes in the ucOCN treated group. The above results suggested that ucOCN inhibited the motility of RAW264.7 cells.

### ucOCN suppressed RANKL-induced osteoclastogenesis

ucOCN affected the osteoclastic gene expression. To clarify the effects of ucOCN on osteoclastogenesis, we examined the osteoclastic gene expression during induction in the presence of ucOCN by RT-qPCR. The preosteoclastic RAW264.7 cells were induced to osteoclastic differentiation by 50 ng/ml receptor activator of nuclear factor-$\kappa$B

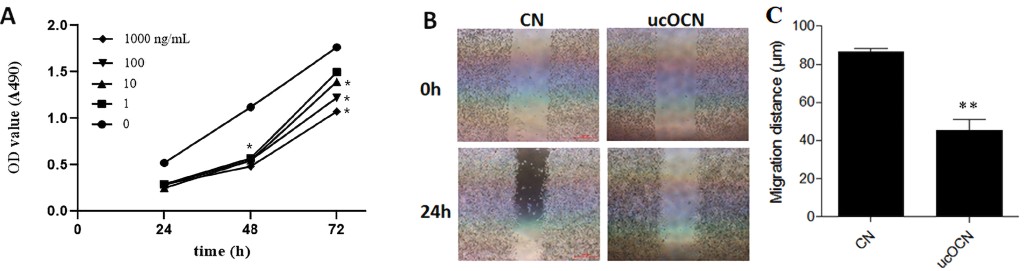

**Figure 1** **The effects of ucOCN on the proliferative activity and migration of RAW264.7.** Cells were treated with ucOCN for different times and measured by MTS for its proliferation (A). The migration of RAW264.7 cells were measured by a scrape wounding assay over 24 h (B) and its corresponding distance statistical results (C). * $P < 0.05$, ** $P < 0.01$, $n \geq 3$, vs. 0 ng/ml group or CN.

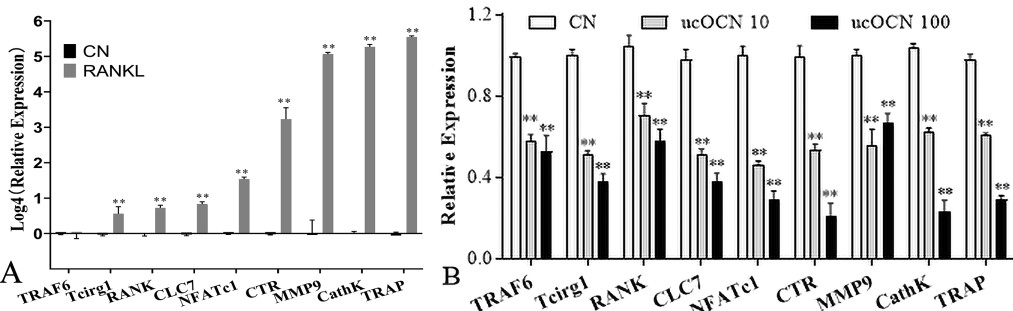

**Figure 2** **ucOCN inhibited osteoclast-related gene expression of RAW264.7 cells induced by RANKL.** RANKL successfully induced the increased expression of osteoclast genes (A), which was significantly attenuated by ucOCN treatment (B). * $P < 0.05$, ** $P < 0.01$, $n \geq 3$ vs. CN (the ucOCN untreated control group).

ligand (RANKL), a key osteoclastogenic cytokine, without or with ucOCN (10 ng/ml or 100 ng/ml based on the proliferation effects results) for 4 days, then lysed by Trizol for gene expression assay. As shown in Fig. 2A, the marker genes (NFATc1, RANK, TRAF6, etc.) of osteoclastic differentiation were significantly increased following RANKL induction, especially the upregulation of CathK and TRAP genes about several hundred folds. Compared with the ucnOCN untreated control group (CN) coupled with RANKL induction, the expression of these genes dramatically decreased about 40–60% after treatment with ucOCN accompanied with RANKL induction. The inhibition efficiency of 100 ng/ml ucOCN was higher than that of 10 ng/ml (Fig. 2B) and it was reported that serum concentration of ucOCN could be 35.9 ng/mL in mice (*Prats-Puig et al., 2014*), So 100 ng/ml was used in the following experiments.

Secondarily, mature TRAP positive multinucleated cells (MNCs) were assayed in RAW276.4 cells and BMMs. Cells were incubated with 100 ng/mL ucOCN in the presence of RANKL for RAW264.7 cells or in the presence of RANKL and MCSF for BMMs. RAW264.7 cells differentiated to mature TRAP-positive MNCs in the absence of ucOCN, however, fewer MNCs formation was observed in the presence of 100ng/ml ucOCN (Fig. 3A).

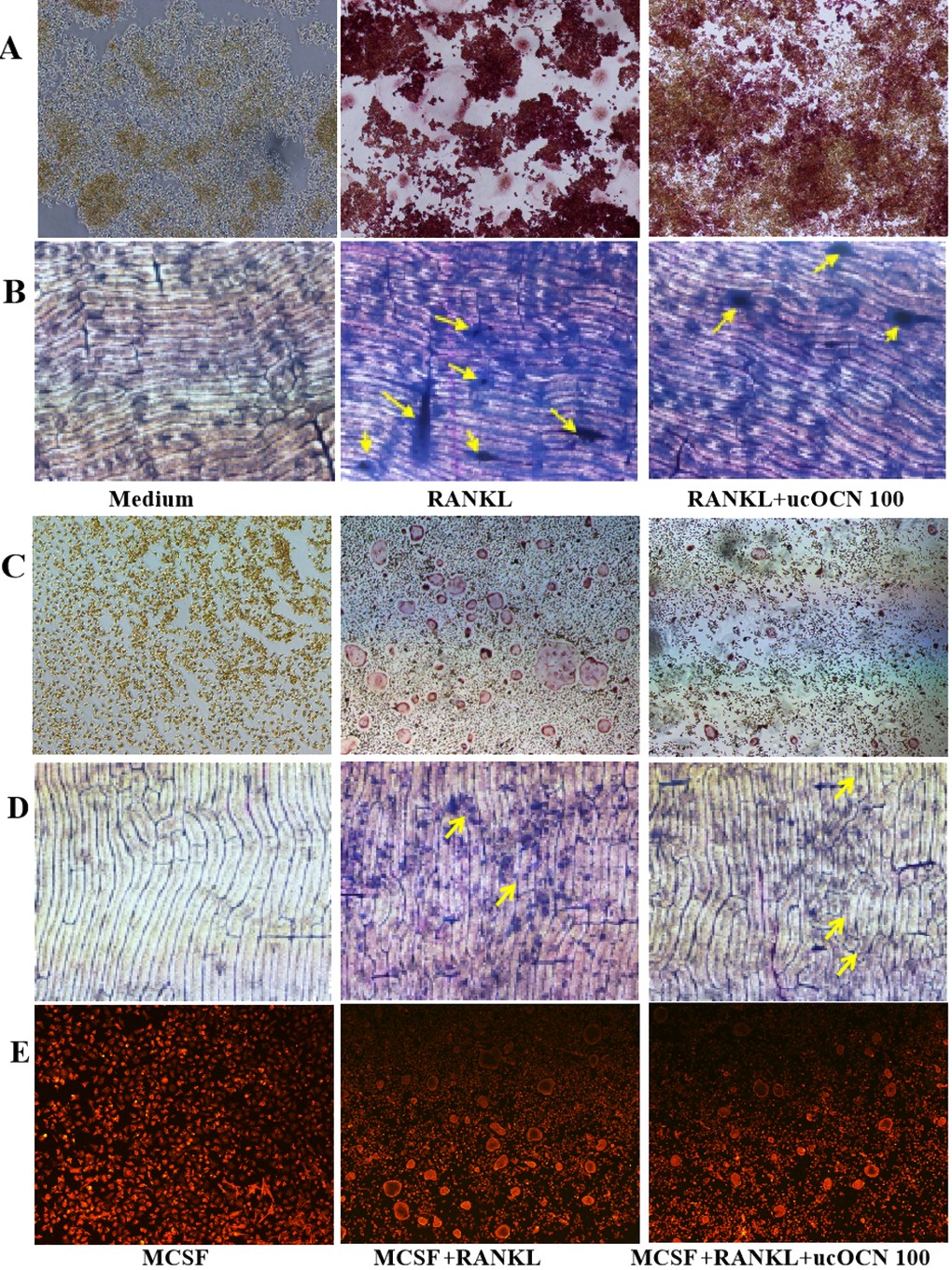

**Figure 3   ucOCN decreased the TRAP positive MNCs, bone resorption activity and actin ring forma-tion.** The first column was the negative control. Cells were incubated with ucOCN (100 ng/ml) in the presence of RANKL (50 ng/ml) for RAW264.7 cells or in the presence of RANKL(50 ng/ml) and MCSF (30 ng/ml) for BMMs for 4–6 days, then performed for TRAP staining (4 days) of RAW264.7 cells (A) and BMMs (C). Toluidine blue staining (6 days) of RAW264.7 cells (B) and BMMs (D). Actin ring staining of differentiated BMMs (E) Image was taken by fluorescence microscope with a 4× object lens.

ucOCN inhibited the formation and number of TRAP positive MNCs. In BMMs, ucOCN appeared to have similar effects. Indeed, there were no purple TRAP positive cells after treatment of MCSF or combined with ucOCN, neither mononuclear cells or multinucleated cells (Fig. S2). The treatment of BMMs with RANKL and MCSF induced the formation of TRAP positive MNCs, nevertheless ucOCN decreased the induction effects of RANKL to osteoclast (Fig. 3C).

We further examined the effects of ucOCN on the function of mature osteoclasts to resorb bone. RAW264.7 cells were plated onto bone slices and induced with RANKL in the presence of 100 ng/mL ucOCN. RANKL stimulated cells form a number of pits, which indicated successful induction of RAW264.7 cells to functional bone resorption activity by RANKL. As shown in Fig. 3B, treatment of 100 ng/ml ucOCN significantly reduced the formation of resorption pits in number and areas compared with treatment with RANKL alone. Similar results were observed in BMMs after induced with MCSF+RANKL combined with ucOCN (Fig. 3D). During bone resorption, an actin ring was formed to seal the resorption area. To investigate the effects of ucOCN on actin ring formation, immunofluorescence staining was carried out for BMM cells induced by RANKL in the presence of ucOCN. There were single cells stained with red in only MCSF treatment group. Well-formed actin rings were observed in the MCSF and RANKL-treated cells. After the addition of ucOCN during the progress of induction of MCSF and RANKL, the number of red actin rings decreased, and the structure of formed actin rings showed irregularity and shrinkage (Fig. 3E). These results indicated the inhibitory effects of ucOCN on the bone resorption of osteoclast and the formation and integrity of the osteoclast actin ring.

## ucOCN affected the initial differentiation of osteoclasts

The above results suggested the inhibition effects of ucOCN on osteoclast differentiation. Next, we tried to explore which stage of osteoclast differentiation was affected by ucOCN. To achieve this aim, ucOCN was added to the MCSF+RANKL inducing medium in the final concentration of 100 ng/mL for BMM cells at different time points as indicated (0, 2nd, 4th day) in Fig. 4A, then some classic osteoclastic genes were detected by RT-qPCR. MNCs were assayed by TRAP staining at the 6th day. Compared with the ucOCN untreated group (CN), the expressions of RANK, TRAP, CathK, and CTR were significantly decreased about 20–50% when ucOCN was simultaneously added with inducing reagent, and only descent tendency was observed when ucOCN was added at the 2nd day or almost no effects were detected when ucOCN was added at the 4th day (Fig. 4B). The inhibition ratio of ucOCN on the expression of classic osteoclastic genes at 0 day (20–50%) was similar to that of RAW264.7 cells as shown in Fig. 2B (40–60%). There was abundant of TRAP positive MNCs in the MCSF and RANKL inducing group, which was markedly reduced in the ucOCN additional group at day 0. The number of TRAP positive MNCs showed a slight decreased in the 2nd day group and no changes occurred in the 4th day group when compared with the ucOCN untreated inducing group (Fig. 4C). It was accordant with the results of gene expression. These results demonstrated that ucOCN could attenuated the early differentiation of osteoclasts.

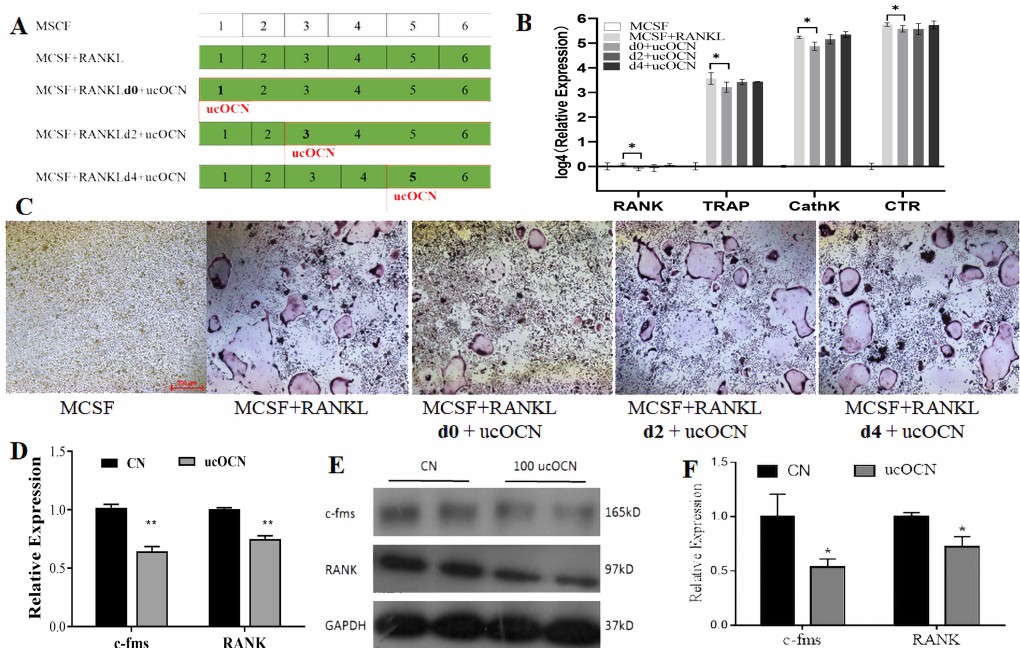

**Figure 4** **ucOCN inhibited the early differentiation of BMMs.** Experimental design and operation chart (A). RT-qPCR detected osteoclast-relate gene expression (B). TRAP staining (C). RAW264.7 cells were treated with ucOCN for 2 day without inducing factor RANKL, then performed RT-qPCR detection for the mRNA expression (D), Western blot detection for the protein expression (E) of c-Fms and RANK and its gray analysis of protein bands (F). $*$ $P < 0.05$, $**$ $P < 0.01$, $n \geq 3$, vs. CN (the ucOCN untreated control group).

To further confirmation this phenomenon, RAW264.7 cells were added with 100 ng/mL ucOCN for 2 days, then evaluated the expression of c-Fms (the receptor of MCSF) and RANK (the receptor of RANKL) genes by RT-qPCR, western blot. As shown in Figs. 4D–4F, the mRNA and protein expression of c-Fms and RANK were inhibited about 30–50% after addition of ucOCN. These results suggested that ucOCN could inhibit the expression of RANK and c-Fms in RAW264.7 cells.

## Gprc6a mediated the effects of ucOCN on osteoclasts

ucOCN promoted beta-cells proliferation and testosterone synthesis in Leydig by Gprc6a, an orphan G protein-coupled receptor (*Oury et al., 2011*; *Wei et al., 2014*). Gprc6a is a cation-, calcimimetic-, and OCN-sensing receptor and widely expressed in brain and peripheral tissues, with the highest levels in kidney, skeletal muscle, testis and leucocyte (*Wellendorph & Brauner-Osborne, 2004*). but it was unknown if osteoclastic lineage cells expressed Gprc6a. In this regard, The expression of Gprc6a in RAW264.7 cells and BMMs were detected by RT-PCR, western blot, and immunofluorescent staining method. Similar to the positive control TC-6 cells, bands of the desired size were observed in the agarose gel electrophoresis of RT-PCR production and in the PVDF film of protein by western blot detection from both osteoclastic lineage cells (Figs. 5A and 5B). Immunofluorescent

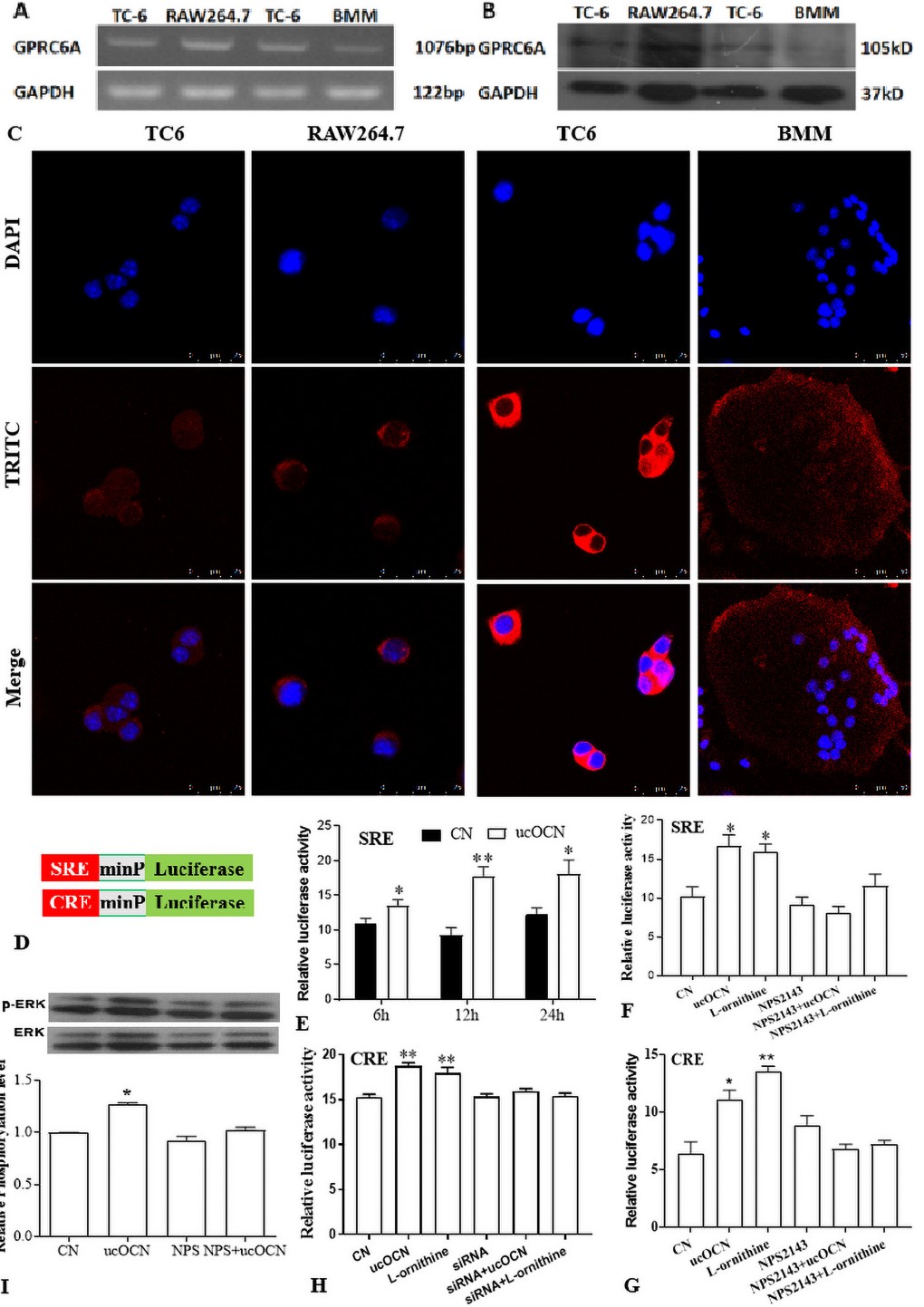

**Figure 5   Gprc6a was expressed in osteoclast and mediated the effects of ucOCN on osteoclast.** mRNA (A) and protein (B) expression detection in TC-6(beta cell as positive control), RAW264.7 cells and BMMs. immunofluorescent staining results of Gprc6a in RAW264.7 cells and mature osteoclast from BMMs (C). SRE- and CRE-miniP (minimal promoter)-luciferase reporter assay system (D) detected the SRE activity in different time with ucOCN addition (E). 

**Figure 5 (…continued)**
SRE (F) and CRE (G) activity after treatment ucOCN for 12 after 36 h reporter transfection in RAW264.7 cells. Gprc6a siRNA was transfected into RAW264.7 cells for 48 h before CRE luciferase transfection, ucOCN treatment, and reporter detection as above description(H). Western blot detection of the phosphorylated ERK 12 h after ucOCN treatment and its band gray analysis (I). * $P < 0.05$, ** $P < 0.01$, $n \geq 3$, vs. CN (the control group without treatments).

staining also demonstrated the expression of Gprc6a, which localized in the cytoplasm membrane of RAW264.7 cells and mature osteoclast (Fig. 5C).

Several reports have demonstrated that ucOCN could activate ERK and serum response element (SRE)-luciferase reporter activity (*Pi et al., 2005*), a transcriptional effector CREB in Leydig (*Oury et al., 2011*). To evaluate if Gprc6a was activated by ucOCN in osteoclastic linage cells, the SRE- or CRE- (cAMP response element) luciferase reporter was transfected into RAW264.7 cells coupled with Renilla phRLTK vector. After transfection for 36 h, the cells were treated with ucOCN, L-ornithine (an agonist of Gprc6a) with or without 10 µM NPS2143 (non-competitive antagonist of Gprc6a (*Faure et al., 2009*)), and then the transcriptional activity was measured with the Dual Luciferase Assay. To screen an optimal time point, the cells were treated for 6 h, 12 h, and 24 h after 36 h of reporter transfection, as illustrated in Fig. 5D. The luciferase activity was significantly increased after treatment with ucOCN at the indicated time, especially at 12 h which was performed in the following experiment (Fig. 5E). Although there was no effect of NPS2143 on the luciferase activity, the promotive effects were eliminated by NPS2143 co-treatment, which was concordance with the positive control groups of L-ornithine (Fig. 5F). Similar results were achieved by CRE-luciferase detection (Fig. 5G). Moreover, increased ERK1/2 phosphorylation was observed after 12 h treatment of ucOCN or L-ornithine, which was also wiped out by NPS2143 (Fig. 5I). Next, a Gprc6a specific siRNA (*Pi et al., 2010*) was transfected into RAW267.4 cells for 48 h, followed by CRE-luciferase reporter transfection and ucOCN treatment as the above description. As shown in Fig. 5H, the promotion of ucOCN and L-ornithine to CRE luciferase activity was completely ablated by the siRNA. These results implied that ucOCN promoted SRE/CRE signaling and ERK1/2 phosphorylation mediated by Gprc6a.

## DISCUSSION

Currently, more and more evidences indicated that ucOCN is an endocrine hormone in regulating energy metabolism, brain development, muscle function, and male male fertility, but some discrepancies and paradoxes emerged in the researches about the structure, physiological function, receptor and clinical effects of OCN (*Li et al., 2016*; *Moriishi et al., 2020*). Early reports have demonstrated that OCN promoted the recruitment and differentiation of osteoclasts, but absent of OCN didn't impair the bone resorption using the ovariectomy model. Recently, Moriishi et al. demonstrate that OCN is not involved in the regulation of bone quantity, glucose metabolism, but required for bone quality and strength by adjusting the alignment of bone apatite crystallites parallel to collagen fibrils (*Moriishi et al., 2020*). Till now, the exact function and mechanism of OCN in bone

remodeling are not well understood. In the present study, we demonstrated that ucOCN inhibited the migration, proliferation, and differentiation of osteoclasts in vitro.

To our knowledge, these results firstly showed that ucOCN had significant inhibitory effects on the proliferation, migration, differentiation, and maturation of osteoclasts in both osteoclast models (RAW264.7 cells and BMMs). First of all, ucOCN attenuated the proliferation and motility of RAW264.7 cells with concentration and time-dependant pattern by MTS detection and scraped wounding assay (Fig. 1). Secondary, osteoclast-related gene expression during osteoclast induction differentiation by RANKL was inhibited by ucOCN, including the master transcription factor NFATc1 (*Kim & Kim, 2014*) and later functional gene TRAP and CathK as shown in Fig. 2. Thirdly, TRAP staining and actin ring staining showed that ucOCN weakened the formation of osteoclast multinuclear fusion. Meanwhile, bone resorption activity was attenuated by ucOCN using toluidine blue staining of bovine bone slices (Fig. 3). Interestingly, we found the inhibitory effects mainly occurred in the initial stage of RANKL-inducing osteoclast differentiation. As shown in Figs. 4D–4F, the early regulation gene c-Fms and RANK were markedly decreased after ucOCN treatment. Finally, the reporter assay system and siRNA silence experiments demonstrated Gprc6a mediated effects of ucOCN on osteoclasts (Fig. 5). All these data demonstrated that ucOCN had a significant inhibitory effect on the differentiation and maturation of osteoclasts and bone resorption.

Bone reconstruction with bone formation by osteoblast and bone resorption by osteoclast ensured the continuous renewal of the body skeletal and maintained its homeostasis. The chemoattractants responsible for the mobilization of osteoclast precursors were derived from the bone matrix (*Malone et al., 1982*). OCN was mainly produced by osteoblasts with its carboxylated form and stored in the bone matrix (*Al Rifai et al., 2017*; *Hauschka et al., 1989*). Osteoblast also released some ucOCN because of the decrease of GGCX activity or insufficiency of vitamin K supplement (*Shiba et al., 2014*). OCN-deficiency BP recruited fewer osteoclast progenitors and formed osteoclasts with decreased TRAP activity (*Glowacki et al., 1991*). OCN fragments, purified from bovine bone, influenced osteoclast maturation, especially in the late stage of osteoclast differentiation (*Ishida & Amano, 2004*). No abnormalities in bone size, morphology, and mineralization were observed in the warfarin-maintained rat bones with only 2% of control OCN levels (*Price & Williamson, 1981*). However, another study showed that rats maintained for 8 months with warfarin were characterized by complete fusion of the proximal tibial growth and cessation of longitudinal growth, which might be caused by the decrease of bone OCN (*Price et al., 1982*). The OCN used in the early studies in the 1980s and 1990s was extracted from animal bones by EDTA method, which included the main component of cOCN and contained some undercarboxylated and fragment OCN. Our results suggested that ucOCN had some negative effects on osteoclasts, which implied the promotion effects of OCN might mainly come from cOCN. Recently, Ferron reported that the osteoblast specific knockout of GGCX had no effect on bone mass and bone turnover parameters (*Ferron et al., 2015*). The GGCX specific knockout or warfarin treatment also decreased the OCN content in bone tissue, which might attenuate the promotion of cOCN on osteoclasts. The low pH in the resorption lacuna could decarboxylate the deposited cOCN and ucOCN

was released into the bloodstream (*Lacombe, Karsenty & Ferron, 2013*). Combined with previous reports about the positive effects of cOCN on the chemotaxis, adhesion, and differentiation of osteoclasts, it was presumed that the balance between carboxylated and undercarboxylated OCN is important for bone metabolism homeostasis. This inhibition effect may also act as a brake for the energy regulation endocrine function of bone. As previous reports, ucOCN increased beta-cell proliferation and insulin secretion (*Lee et al., 2007*). In turn, insulin signaling in osteoblast promoted the osteoclast ability to acidify the bone extracellular matrix, which decarboxylated cOCN into ucOCN and favored whole body glucose homeostasis (*Ferron et al., 2010*). Bone-derived ucOCN and pancreatic $\beta$-cells insulin formed a positive feed-forward regulation loop in glucose regulation. Our results may provide a negative regulation of this cross talk between OCN and insulin, like its negative regulator DLK1 in pancreatic $\beta$-cells (*Abdallah et al., 2015*).

Several studies had shown that the beneficial effects of ucOCN were ameliorated or absent in Gprc6a KO mice, which implicated the class C GPCR (Gprc6a) as a target receptor for ucOCN (*Pi, Wu & Quarles, 2011*), but other KO mice displayed a subtler phenotype with no evidence of impaired glucose handling and insulin sensitivity (*Jorgensen et al., 2019*; *Smajilovic et al., 2013*). Moreover, *Rueda et al. (2016)* showed evidence that Gprc6a was not a direct ligand for ucOCN, rather than an indirect requirement for the effects of OCN. L-amino-sensing, activation of cAMP signaling and ERK1/2 phosphorylation properties of Gprc6a were well characterized, so does the SRE activity. Small molecules, such as NPS-2143, which could bind to the 7TM domain of the receptor, acted as negative allosteric modulators of Gprc6a (*Faure et al., 2009*). Similar to L-ornithine, ucOCN treatment significantly increased the SRE- or CRE-luciferase activity, and ERK1/2 phosphorylation in a NPS2143-sensitive manner. Downregulation of Gprc6a by specific siRNA also attenuated the promotion of ucOCN on CRE-luciferase. These results indicated a Gprc6a-mediated mechanism of ucOCN effects on osteoclasts (Fig. 5).

The downstream signaling of Gprc6a on the inhibition of osteoclasts by ucOCN should be further explored. Our results showed the increased ERK1/2 phosphorylation, which was reported to be a positive regulation on osteoclast mature (*Soysa et al., 2012*), and early reports supported that the MEK/ERK pathway negatively regulates osteoclastogenesis, while the p38 pathway does so positively (*Hotokezaka et al., 2002*). By the way, the role of ucOCN on osteoclasts was similar to the hormone calcitonin, which stimulated cAMP production and inhibited osteoclastic bone resorption (*Yang & Kream, 2008*).

## CONCLUSIONS

Taken together, we firstly provided evidences that ucOCN had negative effects on the proliferation, migration, early differentiation, and bone resorption of osteoclast, which was mediated by Gprc6a. This functional pattern could sever as a negative regulation mechanism for serious bone resorption during bone remodeling and for the feed-forward loop of OCN-insulin on whole body glucose homeostasis and presumed that the balance between carboxylated and undercarboxylated OCN might play an important role in bone remodeling, but the detail mechanism should be further explored. The increase of ucOCN

percentage during spaceflight maybe a negative feedback to prevent the further bone resorption. For the difficult realization by current technology, our notion was a deficiency of evidence from in vivo experiments.

### Funding
This work was supported by the National Natural Science Foundation of China (NO. 31470832, 81801872); Advanced Space Medico-Engineering Research Project of China (18035020103); the State Key Laboratory of Space Medicine Fundamentals and Application and the China Astronaut Research and Training Center (SMFA17A02, SMFA17B04, SMFA17B06, SMFA18B02). The funders had no role in study design, data collection and analysis, decision to publish, or preparation of the manuscript.

### Grant Disclosures
The following grant information was disclosed by the authors:
National Natural Science Foundation of China: 31470832, 81801872.
Advanced Space Medico-Engineering Research Project of China: 18035020103.
State Key Laboratory of Space Medicine Fundamentals and Application.
China Astronaut Research and Training Center: SMFA17A02, SMFA17B04, SMFA17B06, SMFA18B02.

### Competing Interests
The authors declare there are no competing interests.

### Author Contributions
- Hailong Wang and Hongyu Zhang performed the experiments, analyzed the data, authored or reviewed drafts of the paper, and approved the final draft.
- Jinqiao Li performed the experiments, analyzed the data, prepared figures and/or tables, authored or reviewed drafts of the paper, and approved the final draft.
- Zihan Xu, Feng Wu, Chao Yang and Jian Chen performed the experiments, analyzed the data, prepared figures and/or tables, and approved the final draft.
- Bai Ding and Zhongquan Dai conceived and designed the experiments, analyzed the data, prepared figures and/or tables, authored or reviewed drafts of the paper, and approved the final draft.
- Xiukun Sui analyzed the data, prepared figures and/or tables, authored or reviewed drafts of the paper, and approved the final draft.
- Zhifeng Guo performed the experiments, prepared figures and/or tables, and approved the final draft.
- Yinghui Li conceived and designed the experiments, authored or reviewed drafts of the paper, and approved the final draft.

### Data Availability
The raw measurements are available in the Supplemental Files.

## Supplemental Information

Supplemental information for this article can be found online at http://dx.doi.org/10.7717/peerj.10898#supplemental-information.

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
