# Peer review of "Undercarboxylated osteocalcin inhibits the early differentiation of osteoclast mediated by Gprc6a"

_PeerJ, doi:10.7717/peerj.10898_

## Round 0.1 · original submission · Major Revisions

Two experts have reviewed, and they find that your paper is novel and excellent. However, they also comment on a lot of points to be revised. I think those comments are constructive and therefore necessary to improve the paper.

Reviewer 1 ·

Basic reporting

This article is clearly written and includes all the relevant papers are cited

Experimental design

This manuscript presents some level of novelty. Scientific question is well defined and methodology is described.

Validity of the findings

If the recombinant osteocalcin used in this study is sufficient, conclusions are supported with the results. Statistic is adequate.

Additional comments

This manuscript investigates the effect uncarboxylated osteocalcin on RAW264.7-derived osteoclasts. Overall, this study is well conducted and the experiments are correctly designed Also this study adds some novel aspects which were not investigated before. However, one major concern, which should be addressed before the publication is the purity of the osteocalcin used in this study. The authors generated recombinant osteocalcin using bacterial expression system and measured its concentration using specific ELISA. To draw the conclusion that the effects observed in this study are related to osteocalcin the authors need to prove that their recombinant protein do not contain contaminants of other proteins or components of the bacteria. They should perform an SDS page with purified osteocalcin followed by coomassie blue staining. They should also test the solution with recombinant osteocalcin for any bacterial components.

Reviewer 2 ·

Basic reporting

I found the text poorly written and difficult to follow. Major revisions of the written English are required.
Figure 1A is difficult to interpret. For me is much more intuitive to plot a time vs OD curve, distinguishing the different concentrations with different curves.
Line 61 to 66. This sentence is completely incomprehensible to me. Why should OCN have a role, either carboxylated or not, in OCN-deficient bone particles (BP) resorption? What are bone particles? Where are they found? What is their pathophysiological significance?

Experimental design

The overall experimental session lack of a comparison between carboxylated OCN and ucOCN, being mainly focused on the latter. This is a key point to understand the physiological role of the inhibitory effect of ucOCN on Ocl. In other words, why should Ocl be inhibited by the product of their activity? (considering that, at least in mouse, bone resorption is partially stimulated by osteoblasts through the action if insulin).

Validity of the findings

The choice of 100 ng/mL of ucOCN to investigate the mediation of GPRC6A is questionable since the average serum concentration of ucOCN is about 10 ng/mL. Are there any studies reporting bone concentration of ucOCN greater than the serum?

Additional comments

In this paper, Wang et al. investigate the possible inhibitory activity of undercarboxylated osteocalcin (ucOCN) on osteoclast (Ocl)differentiation and function. Authors find that ucOCN inhibits Ocl precursors proliferation, migration and differentiation into mature Ocl. In addition, ucOCN seems to inhibit the bone remodeling function of this cell population. All these effects are mediated by GPRC6A.
It is a quite interesting study, disclosing some novel effects of osteocalcin in its main tissue of expression. However, I have many concerns that require to be addressed.

---

## Round 0.2 · Minor Revisions

A few minor changes are still needed:

Fig. 1A does not include the OD at the initial time point: this makes it impossible to understand if the changes observed upon ucOCN incubation are completely due to ucOCN or to heterogeneities in the initial ODs.

The legends to Figures 2-5 contains an undefined abbreviation (CN). Sub-par language is still present. Some examples include: line 193 " The inhibition of the lowest concentration (1 ng/ml) just lasted for 48 h, but the other three concentrations continuously occurred for 72 h." or line 219 ". Cells were incubated with ucOCN as indicated concentration".

Reviewer 1 ·

Basic reporting

This article is clearly written and includes all the relevant papers are cited

Experimental design

This manuscript presents some level of novelty. The scientific question is well defined and methodology is described.

Validity of the findings

Conclusions are supported by the results. The statistic is adequate.

Additional comments

All of my concerns have been addressed

---

## Round 0.3 · accepted · Accept

Thank you for addressing the remaining issues and revising the manuscript.